# Dry or Wet? Evaluating the Initial Rice Cultivation Environment on the Korean Peninsula

**Shinya Shoda** [1,2,*] **, Hiroo Nasu** [3] **, Kohei Yamazaki** [4] **, Natsuki Murakami** [1] **, Geon-Ju Na** [5] **, Sung-Mo Ahn** [6] **and Minoru Yoneda** [4]

1   International Cooperation Section, Nara National Research Institute for Cultural Properties, Nijo 2-9-1, Nara 630-8577, Japan; murakami-n33@nich.go.jp
2   BioArCh, Department of Archaeology, University of York, Wentworth Way, Heslington, York YO10 5NG, UK
3   Center for Fundamental Education, Okayama University of Science, 1-1 Ridaicho, Kita Ward, Okayama 700-0005, Japan; nasu@big.ous.ac.jp
4   The University Museum, The University of Tokyo, Hongo 7-3-1, Bunkyo, Tokyo 113-0033, Japan; yamazaki@um.u-tokyo.ac.jp (K.Y.); myoneda@um.u-tokyo.ac.jp (M.Y.)
5   Geumgang Research Institute for Cultural Heritage, 155-1 Deogmyeong-dong, Yuseong-gu, Daejeon 34154, Korea; sobakhan@hanmail.net
6   College of Humanity, Wonkwang University, 460 Iksan-daero, Iksan-si 54538, Jeollabuk-do, Korea; sungmo@wku.ac.kr
*   Correspondence: shinya.shoda@york.ac.uk; Tel.: +81-742-30-6839

**Abstract:** The origins and development of rice cultivation are one of the most important aspects in studying agricultural and socio-economic innovations, as well as environmental change, in East Asian prehistory. In particular, whether wet or dry rice cultivation was conducted is an important consideration of its impact on societies and the environment across different periods and places. In this study, carbon and nitrogen stable isotope analysis of charred crop remains from archaeological sites dating from the Early Bronze Age (ca. 1.1 k BC) to the Proto-Three Kingdoms (ca. 0.4 k AD) was conducted to clarify: (1) if there were any shifts from dry to wet cultivation around 1500 years after rice adoption as previously hypothesized and (2) the difference in stable carbon and nitrogen isotope values between rice and dry fields crops excavated from the same archaeological context to understand the cultivation environment. The result show that stable isotope values of charred rice grains have not changed significantly for around 1500 years. Moreover, rice possessed higher nitrogen stable isotope values than dry crops across all periods. While other potential factors could have influenced the $^{15}$N-enrichment of soils and crops, the most reasonable explanation is bacteriologic denitrification in anaerobic paddy soil where the rice was grown.

**Keywords:** stable isotopes; paleoethnobotany; crops; Korean Bronze Age; Proto-Three Kingdom period

## 1. Introduction

Rice (*Oryza sativa* L.) is undoubtedly one of the most important crops in modern societies. The significance of studying the origins and development of rice cultivation is widely recognized because it has played a crucial role in the development of early agricultural societies and environmental change in East Asia [1,2]. Korea is not an exception, where rice has been cultivated since the second millennium BC [3,4]. However, the available archaeological lines of evidence are not sufficient to determine whether the initial form of rice cultivation was wet or dry despite its importance to understand the development of rice production systems and their role in both socio-economic and natural environmental impacts—such as methane expansion, which causes global warming [2,5,6]. Despite the lack of evidence, some archaeologists have assumed that societal change within the Korean Bronze Age corresponded to the transition from dry to wet rice cultivation [7].

Since rice is the only crop that could have been cultivated in a wet environment among the major East Asian crops, we investigated whether there is a significant difference in plant

nitrogen stable isotope values ($\delta^{15}$N) between rice and these other crops—which can be explained by bacteriologic denitrification in anaerobic paddy soil, which could affect $\delta^{15}$N values [8]. To date, this has not been previously investigated for plant remains recovered from Korean archaeological sites. $\delta^{15}$N values were investigated in some of the earliest rice grains of the Early Bronze Age (hereafter EBA). For comparison, remains dating to the Late Bronze Age (hereafter LBA) and the Proto-Three Kingdom period (hereafter PTK) were analyzed. Other charred plant remains, including C3 crops (wheat), C4 crops (foxtail millet), and $N_2$-fixing legumes (azuki bean) were additionally sampled to estimate the impact of diagenesis and reconstruct the prehistoric plant management.

The origins of rice domestication can be traced back to the middle and lower Yangtze River basin in the mid-southern part of China [9–11]. Here, there was a natural habitat of wild rice [12]. Rice cultivation was developed within water-managed systems, represented by paddy fields [13]. In a recent study on rice phytolith morphology, changes in the arable system of the lower Yangtze region were identified. In that study, rice was grown in a similar habitat to wild rice at ca. 4.8–4.3 k cal BC, which then moved into a controlled water drainage system by ca. 1.9–1.7 k cal BC, and finally within highly irrigated paddy fields by ca. 1.0–0.3 k BC [2]. Although there are some disagreements on the timing of its domestication [14–16], rice seems to have spread into the Shandong peninsula, adjacent to the Korean peninsula, as early as 7050 ± 80 BP (Before Present) according to $^{14}$C dating (ca. 6.1–5.7 k cal BC) [17].

As for the Korean peninsula, rice seems to have been introduced around the beginning of the Bronze Age (or Mumun period), dated to the final quarter of the second millennium BC [3,4]—about two thousand years later than the introduction of millet [18]. There has been some debate about the "oldest" rice. "Paleolithic rice" was reported from the Sorori site where rice husks were found within soil dated between 16 and 13 k cal BC [19]. However, other researchers reported that the rice grain was $^{14}$C dated to 12,520 ± 150 BP (ca. 13.3–12.2 k cal BC) while the surrounding peat was $^{14}$C dated to 12,552 ± 90 BP (ca. 13.2–12.4 cal BC), reconfirming an old date for the Sorori rice [20]. However, the majority of samples (six out of seven) were found to be modern, and the difference in dates from the previous study was not explained. In addition, "Neolithic rice" was reported from the Daecheon-ri site, where charred rice grains were recovered from a dwelling pit dated to ca. 3.5–2.5 k cal BC, based on the charcoal sample [21]. Recently, an AMS (Accelerator Mass Spectrometry) date revealed that it was $^{14}$C dated to as late as 2070 ± 60 BP (ca. 0.3 k cal BC–0.4 k cal AD) [22]. In short, there is no reliable evidence for rice dating back to either the Paleolithic or Neolithic period (or Chulmun period) on the Korean peninsula [3,4].

Traditionally, rice agriculture has been seen as a key driver in the development of complex societies within Korean prehistory [23]. However, archaeological evidence such as paddy fields and a clear inequality in tomb size and offerings, as well as differences in the scale and duration of settlement sites, appeared only in the LBA [24]. This situation has led some archaeologists to believe that rice and other crop cultivation during the EBA developed as a slash-and-burn agriculture, considered "primitive" by some and, supposedly, less influential for social complexity as it requires a smaller scale of social organization than an irrigated paddy field [25]. Subsequently, Ahn claimed that EBA populations practiced slash-and-burn or dry cultivation of rice—based on the stone tool assemblage and the location of settlements. However, the predominance of stone axes and dwelling pits on hills [7] are not necessarily convincing arguments. Even after criticism [26–28], debate remains.

On the other hand, it is generally accepted that rice was cultivated under wet conditions, such as paddy fields, from the LBA, as archaeological paddy fields of corresponding periods have been identified [24], and probably based on the assumption of the continuity of rice cultivation practices into the historic era (and to the present). Annual precipitation in the southern part of today's Korean peninsula is between 500 to 1500 mm which is suitable, or at least acceptable for, maintaining rice paddy fields [29]. According to agricultural records for 1935, most of southern Korea had, in terms of area, more paddy fields than dry fields [30]. The paleoclimatic condition of the study area was investigated [31] and the

periods covered by this paper were within the climate deterioration stage after the end of the Holocene climate optimum. However, it is still not clear how different the precipitation or temperature between the present and the study period was—although it is unlikely that the climate conditions were entirely different.

In general, the hypotheses have posited that prehistoric rice cultivation was divided into two stages: the "primitive" dry field stage and the "developed" wet field stage [25]. These assumptions have two problems, as previously pointed out [26,27]. One is that they consider slash-and-burn cultivation as a "primitive" method of cultivation. There are a variety of ethnographic examples of sophisticated subsistence systems based on slash-and-burn agriculture, and sometimes it supports sedentism rather than mobility—a sedentary society was presumed as the social organization in EBA Korea [28]. In fact, it could produce ecologically sustainable agro-ecosystems under conditions of low land use pressure [32]. Thus, there is no rationale that we consider of this way of cultivation as being primitive or highly mobile.

Additionally, there is no evidence for initial rice cultivation being dry or upland based, needless to mention slash-and-burn cultivation. A predominance of stone axes/adzes [7] does not necessarily equate with the existence of slash-and-burn cultivation since it is not always combined with large-scale logging, but also because these stone tools could be used to produce wooden tools for paddy field management. Settlement location on the hills [7,25] could not support the existence of slash-and-burn as settlements should rather be kept away from the fire stage of the process. There is no reason for the settlements' location (on the hill) to be related to this cultivation style. Thus, it cannot be assumed that slash-and-burn or dry rice cultivation was practiced at the onset of its introduction. In order to understand the development of rice cultivation and agricultural societies, there is a need to distinguish between wet- and dry-farmed rice through the analysis of archaeological remains. This question could be answered by the analysis of weed assemblages [13,33,34], and probably by phytolith morphotypes [2], if there are appropriate materials to study.

However, there is no such sample available for the initial stage of rice cultivation in Korea, as there are no archaeological features that have been identified relating to cultivation. As for the locations of settlement sites, they are mostly located on the hills facing alluvial plains or on the elevated banks and terraces along rivers, which enable residents to conduct both dry and wet rice cultivation. In light of this, another method based on the stable isotope analysis of charred crop grains, not only rice but also dry crops such as foxtail millet, wheat, and azuki beans cultivated at the same time, could be used. This method can be usefully applied to anthropogenic remains, especially charred cereals rather than weeds or phytoliths, which were collected from dwelling pits rather than the fields themselves.

Carbon and nitrogen stable isotope analysis of archaeological cereal remains is a well-known method within archaeobotanical research [35–37]. This method is used for distinguishing harvesting sites or agricultural practices such as manuring or irrigation [38,39], although these studies are mainly concerned with wheat and barley, rather than rice. It is noteworthy that experimental studies have shown that the charring process does not cause a significant change in the $\delta^{13}C$ and $\delta^{15}N$ values for six crop species that have been analyzed (bread wheat, einkorn, emmer, hulled barley, lentil, and pea), which are within 0.5‰ for the former, and within 1‰ for the latter [40].

The variability of $\delta^{15}N$ values in plants is between −8 and +18‰ [41]. Szpak recently summarized the processes leading to the variation of $\delta^{15}N$ in plants as follows: (1) inherent variation in soil; (2) soil N availability; (3) environmental factors such as aridity; and (4) anthropogenic N addition [8]. As for rice, higher $\delta^{15}N$ values are expected within wet rice because large isotopic fractionation through the denitrification by anaerobic bacteria removes $^{15}N$-depleted $N_2$ from paddy soil to the atmosphere [42,43]. While the nitrification–denitrification system makes the $\delta^{15}N$ values in paddy soil variable in relation to the application of fertilizer [44], experimental studies in Japan have shown that wet cultivated rice without the application of chemical fertilizers have higher $\delta^{15}N$ values than dry

upland rice by 2.2–6.1‰, and sorghum by 1.7–6.5‰ [45]. In light of these data, this paper assumes that a similar difference between dry and wet crops is expected from archaeological contexts.

Moreover, if rice was grown in different environments such as water-filled paddy fields or dry fields, the differences in the $\delta^{15}$N values could be observed. Of course, there are a variety of forms of wet rice habitation, from small-scale rain-fed paddy fields to vast deltas and flood plains, but this might be difficult to distinguish using the mentioned method. To test above-mentioned assumption, carbon and nitrogen stable isotope analysis on excavated charred crops covering 1500 years from the beginning of rice cultivation in Korea was conducted to observe if there are any significant changes in their isotope values which might suggest a shift from dry to wet cultivation. The aims of the study were as follows: (1) to clarify if there was any shift of rice $\delta^{15}$N values within 1500 years, which supports the transition from dry to wet cultivation as hypothesized in the previous studies and (2) to clarify the difference in the $\delta^{15}$N and $\delta^{13}$C values between rice and dry field crops to understand the cultivation system.

## 2. Materials and Methods

### 2.1. Charred Grains for the Analaysis

In this study, charred rice grains collected from Korean archaeological sites dating from ca. 1.1 k cal BC to 0.4 k cal AD were analyzed. For comparison, additional species, including foxtail millet (*Setaria italica* (L.) P. Beauv), wheat (*Triticum aestivum* L), and azuki bean (*Vigna angularis* (Willd.) Ohwi et H. Ohashi) grains were sampled. Fifty-two carbonized grain samples were collected from six archaeological sites in South Korea, dated from the end of the second millennium BC (EBA) to the fourth century AD (PTK) to trace the long-term transition of rice cultivation. They were removed directly from the floors of dwelling pits or collected by water flotation of soil from these dwelling pits (for the details of the archaeological sites investigated, see Appendix A).

Firstly, the grain remains were identified to species and grain size measurements were then taken. The carbonized grains were identified using reference collections of modern specimens. The morphological characteristics of each grain are described below (Figure 1). Rice grains are a narrow ellipsoid shape, about 4.3 mm long, 1.8 mm wide, and 2.7 mm thick. In the dorsal and lateral views, traces of the embryo can be seen at the base. There are two longitudinal depressions on the surface of the lateral view. Foxtail millet grains are small and rounded in shape, about 1.3 mm long, 1.3 mm wide, and 1.2 mm thick. The dorsal view shows the remains of an elongated embryo about two-thirds the length of the seed. Wheat grains are ellipsoid in shape, about 3.4 mm long, 2.3 mm wide, and 1.9 mm thick. The dorsal and lateral views show traces of an embryo at the base. In the lateral view, the dorsal surface (embryo side) is rounded and the ventral surface is flat. In the dorsal view, there is a longitudinal groove in the middle. It can be identified as naked wheat because it does not have glumes. There are two types of naked wheat, tetraploid durum wheat (*Triticum durum*) and hexaploid bread wheat (*Triticum aestivum*). However, durum wheat is rarely cultivated in East Asia and is thus most likely bread wheat. Azuki beans are a bale-shaped ellipsoids, about 5.6 mm long, 3.9 mm wide, and 3.7 mm thick. In dorsal and ventral views, there is a longitudinal crack in the center, indicating that it is a cotyledon. In the ventral view, there is a slightly wider depression in the middle, indicating the presence of a hilum. Out of these species, rice is the only one that can be cultivated in paddy fields.

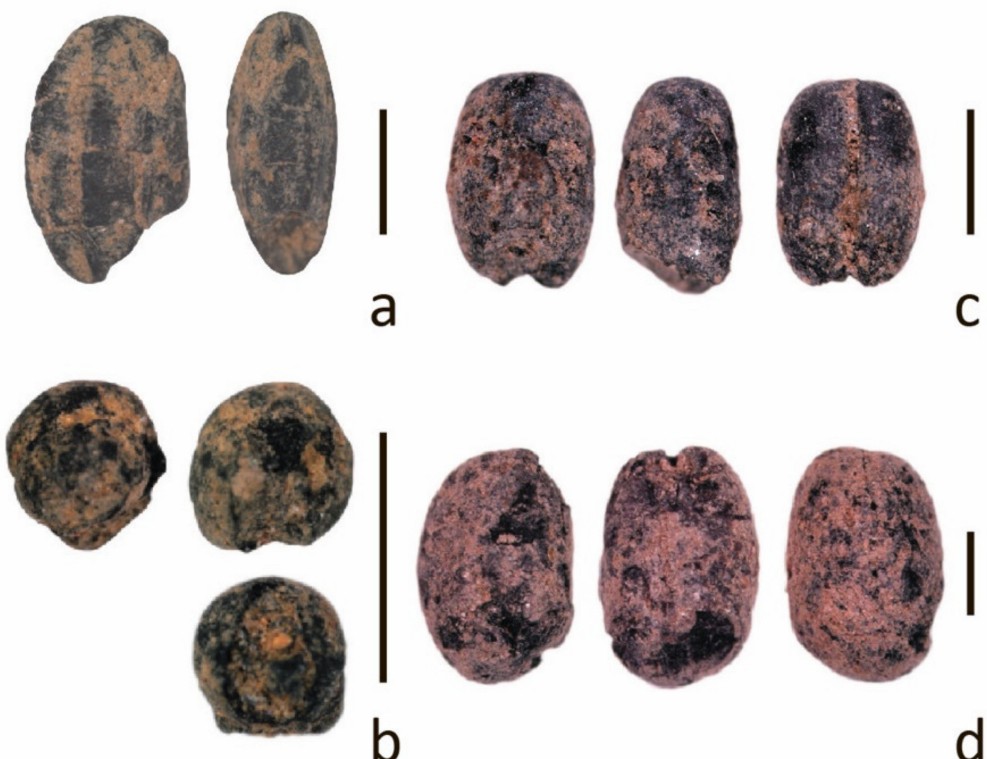

**Figure 1.** Excavated charred grains used for the analysis. A rice (**a**) and a foxtail millet grain (**b**) from dwelling no. 54–2 of the Jagae-ri site and a wheat (**c**) and an azuki bean grain (**d**) from dwelling no. 2 of the Yongheung-ri site. Scale bars = 2 mm.

Secondly, direct AMS radiocarbon dating of charred remains was conducted using Compact AMS 1.5SDH (Nuclear Electrostatics Corporation, Middleton, WI, USA) in the Paleo Labo Ltd. (PED) and CAMS 500 (Nuclear Electrostatics Corporation, Middleton, USA) in the University Museum, the University of Tokyo (MTC) following acid–base–acid (ABA) pre-treatment (HCl: 1.2 N, NaOH: 1.0 N, HCl: 1.2 N). The typological sequence of the materials is presented in Table 1 and Figure 2. For descriptive purposes, EBA was divided into two phases: Gyo-dong as EBA1 (Middle Early Bronze Age) and Sacheon-ri and Baekseok-dong as EBA2 (Late Early Bronze Age)—according to the chronological study of pottery, stone tools, and dwelling pits [24].

**Table 1.** The rice grain samples for AMS dating and their calibrated ages. Calibration was performed using Oxcal v.4.4.2 [46], with atmospheric data by [47].

| Site | Feature | Age/Phase | Sample Nr. | $\delta^{13}$C (‰) | $^{14}$C Age BP | Calibrated Age (1σ) | Calibrated Age (2σ) |
|---|---|---|---|---|---|---|---|
| Gyo-dong | Dwelling no. 1 | Middle EBA (EBA1) | PED-11437 | −26.2 ± 0.11 | 2860 ± 20 | 1103–1100 BC (1.3%) 1077–1071 BC (2.6%) 1055–984 BC (64.3%) | 1115–972 BC (88.0%) 957–932 BC (7.5%) |
| Sacheon-ri | Dwelling C−5 | Late EBA(EBA2) | MTC-13232 | −26.2 ± 0.20 | 2810 ± 150 | 1193–1176 BC (2.5%) 1159–1145 BC (2.3%) 1129–813 BC (63.5%) | 1421–750 BC (93.6%) 685–667 BC (0.5%) 636–588 BC (1.2%) 579–572 BC (0.1%) |
| Baekseok-dong | Dwelling II−10 | Late EBA(EBA2) | PED-16765 | −24.6 ± 0.17 | 2830 ± 20 | 1012–971 BC (43.7%) 957–932 BC (24.6%) | 1048–920 BC (95.4%) |
| Jagae-ri | | Early LBA | | −25.9 ± 0.19 | 2480 ± 20 | 753–725 BC (14.3%) 706–683 BC (11.4%) 668–664 BC (2.2%) 651–632 BC (9.9%) 624–611 BC (5.8%) 592–545 BC (24.7%) | 767–537 BC (94.0%) 530–519 BC (1.5%) |
| Youngheung-ri | Dwelling no. 2 | PTK | PED-11438 | −26.4 ± 0.12 | 1735 ± 20 | 254–287 AD (29.8%) 324–363 AD (38.5%) | 249–402 AD (36.5%) 308–402 AD (58.9%) |
| Simpo-ri | Dwelling no. 1 | PTK | PED-11439 | −25.9 ± 0.18 | 1740 ± 20 | 252–291 AD (32.9%) 320–353 AD (35.4%) | 246–383 AD (95.4%) |

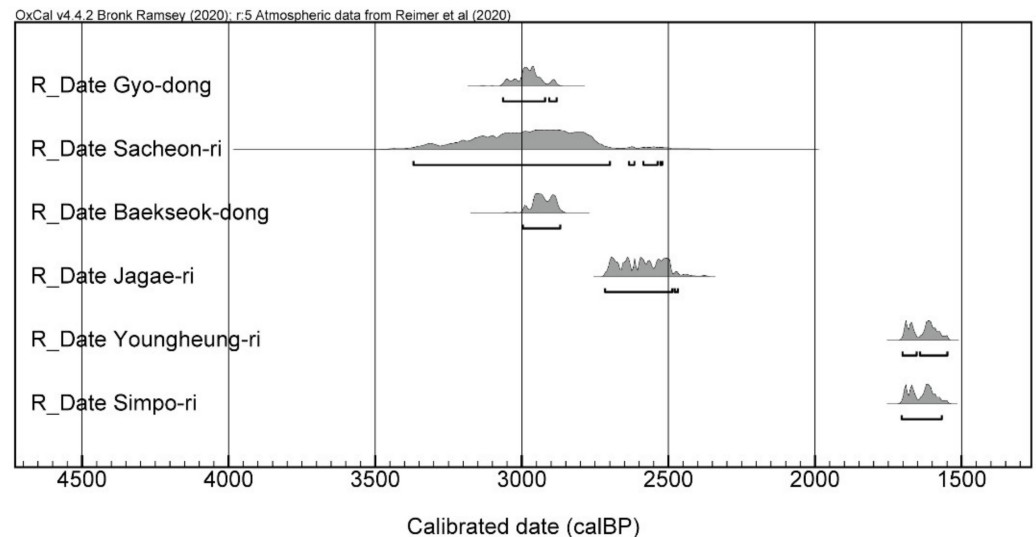

**Figure 2.** Calibrated ages (2σ) of radiocarbon dates of rice grains from the archaeological sites in Korea (see Table 1 for sample information).

## 2.2. Bulk Isotope Analysis of Charred Crop Remains

Carbon and nitrogen stable isotope values ($\delta^{13}$C and $\delta^{15}$N) were determined for the charred crop remains. In addition, international standards (V-PDB and AIR, respectively)

were measured for analytical integrity. The archaeological carbonized grains were crushed (into powder), using an agate pestle and mortar and then they were washed with weak acid (0.1 M HCl). This was followed by ultrasonic cleaning in pure water and lyophilization. Considering that ABA pre-treated samples tend to have higher $\delta^{15}N$ values than non-treated ones [48], alkaline washing was not applied for stable isotope analysis. The samples, weighing around 400 µg, were analyzed to determine the $\delta^{13}C$ values and carbon and nitrogen percentages (%C and %N) by elemental analysis–isotope ratio mass spectrometry (EA-IRMS) using an isotope mass spectrometer (Thermo Delta V) combined with an elemental analyzer (Thermo FLASH2000). Then, the $\delta^{15}N$ values were measured with 90 µg from the aliquot. Typical uncertainty between measurements was estimated to be 0.1‰ for both $\delta^{13}C$ and $\delta^{15}N$ by multiple measurements of internal standards whose isotope values could be traced back to international standards.

### 2.3. Statistics

All statistical calculations were conducted by the statistical programming language R v.4.0.2 [49]. Normality was tested by a Shapiro–Wilk test, which suggested a normal distribution amongst all groups (chemical element/period/species) except for the $\delta^{13}C$ values of rice from EBA1 and EBA2 and the $\delta^{15}N$ values of rice from EBA2. As the distribution of EBA2 was disturbed, due to a sample from Sacheon-ri, the $\delta^{13}C$ and $\delta^{15}N$ from only the Baekseok-dong site ($n = 10$) suggested normal distributions. When homoscedasticity was shown by an F-test in addition to normal distribution, parametric methods were applied to compare averages (*t*-test). Otherwise, non-parametric methods (Mann–Whitney U-test and Kruskal–Wallis test) were also applied when these three methods with biased distributions were used.

## 3. Results

### 3.1. $\delta^{13}C$ in Charred C3 and C4 Crop Grains

While minor differences were found between C3 plants and rice from different periods, the difference between C3 (rice, wheat, and azuki bean) and C4 plants (foxtail millet) was very significant (Table 2, Figure 3), as physiologically expected [50]. In the Baeksoek-dong site, foxtail millet ($-8.9 \pm 0.1‰$; mean and one standard deviation) showed significantly higher $\delta^{13}C$ values than rice ($-25.6 \pm 0.7‰$) (U-test, $p = 0.01398$). Slightly higher $\delta^{13}C$ values in rice from the PTK period ($-24.7 \pm 0.6‰$) could be affected by the fluctuation in $\delta^{13}C$ in atmospheric $CO_2$ as there are a few differences in the $\delta^{13}C$ values of air among the EBA ($-6.52‰$), LBA ($-6.50‰$), and PTK ($-6.36‰$) according to the $\delta^{13}C_{air}$ dataset based on Antarctic ice-core records [51].

**Table 2.** Stable carbon and nitrogen isotope values of the charred crop grains analyzed in this study.

| Site | Age/Phase | Plant Species | Sample Number | %C | %N | $\delta^{13}C$ (‰) | $\delta^{15}N$ (‰) | C/N |
|---|---|---|---|---|---|---|---|---|
| Gyo-dong | Middle EBA (EBA1) | *Oryza sativa* | Gyo-dong−2 | 50.9 | 2.3 | −25.9 | 6.1 | 26.3 |
| Gyo-dong | Middle EBA (EBA1) | *Oryza sativa* | Gyo-dong−3 | 49.9 | 2.2 | −25.9 | 5.6 | 26.2 |
| Gyo-dong | Middle EBA (EBA1) | *Oryza sativa* | Gyo-dong−4 | 54.3 | 1.9 | −25.6 | 5.5 | 33.9 |
| Gyo-dong | Middle EBA (EBA1) | *Oryza sativa* | Gyo-dong−5 | 47.5 | 2.2 | −24.9 | 5.6 | 24.9 |
| Gyo-dong | Middle EBA (EBA1) | *Oryza sativa* | Gyo-dong−6 | 56.1 | 2.1 | −25.7 | 5.9 | 31.0 |
| Gyo-dong | Middle EBA (EBA1) | *Oryza sativa* | Gyo-dong−7 | 44.5 | 2.3 | −25.5 | 6.2 | 22.6 |
| Gyo-dong | Middle EBA (EBA1) | *Oryza sativa* | Gyo-dong−8 | 51.7 | 1.9 | −25.6 | 7.3 | 32.4 |
| Gyo-dong | Middle EBA (EBA1) | *Oryza sativa* | Gyo-dong−9 | 59.3 | 2.5 | −25.7 | 4.9 | 28.2 |
| Gyo-dong | Middle EBA (EBA1) | *Oryza sativa* | Gyo-dong−10 | 54.1 | 1.6 | −25.8 | 4.1 | 39.2 |
| Sacheon-ri | Late EBA (EBA2) | *Oryza sativa* | SCR-r1 | 39.5 | 2.0 | −26.2 | 10.9 | 22.8 |
| Baekseok-dong | Late EBA (EBA2) | *Oryza sativa* | BSGII 10J2 b | 50.4 | 1.7 | −26.2 | 7.3 | 34.7 |
| Baekseok-dong | Late EBA (EBA2) | *Oryza sativa* | BSGII 10J2 c | 47.9 | 2.0 | −26.1 | 7.6 | 28.3 |
| Baekseok-dong | Late EBA (EBA2) | *Oryza sativa* | BSGII 10J2 d | 46.8 | 1.4 | −26.5 | 4.5 | 39.2 |
| Baekseok-dong | Late EBA (EBA2) | *Oryza sativa* | BSGII 10J2 e | 58.2 | 1.9 | −25.4 | 6.6 | 35.6 |
| Baekseok-dong | Late EBA (EBA2) | *Oryza sativa* | BSGII 10J2 f | 53.1 | 1.9 | −24.4 | 6.2 | 32.0 |
| Baekseok-dong | Late EBA (EBA2) | *Oryza sativa* | BSGII 10J2 g | 51.5 | 1.6 | −26.1 | 5.7 | 38.5 |
| Baekseok-dong | Late EBA (EBA2) | *Oryza sativa* | BSGII 10J2 h | 50.1 | 2.0 | −25.8 | 7.2 | 28.6 |
| Baekseok-dong | Late EBA (EBA2) | *Oryza sativa* | BSGII 10J2 i | 49.9 | 1.7 | −24.3 | 6.2 | 35.1 |
| Baekseok-dong | Late EBA (EBA2) | *Oryza sativa* | BSGII 10J2 j | 51.8 | 1.4 | −26.0 | 6.1 | 44.7 |
| Baekseok-dong | Late EBA (EBA2) | *Oryza sativa* | BSGII 10J2 k | 41.1 | 1.2 | −25.5 | 5.7 | 38.8 |
| Baekseok-dong | Late EBA (EBA2) | *Setaria italica* | PTKBSGII 10J2 o/p/q | 51.2 | 2.6 | −9.0 | 5.2 | 23.2 |
| Baekseok-dong | Late EBA (EBA2) | *Setaria italica* | BSGII 10J2 r/s/t | 57.6 | 2.2 | −9.0 | 4.5 | 30.8 |
| Baekseok-dong | Late EBA (EBA2) | *Setaria italica* | BSGII 10J2 u/v/x | 53.3 | 2.2 | −8.8 | 4.6 | 27.8 |
| Jagae-ri | Early LBA | *Oryza sativa* | D48-2 | 45.1 | 2.6 | −25.0 | 7.1 | 20.4 |
| Jagae-ri | Early LBA | *Oryza sativa* | D48-3 | 51.5 | 1.9 | −26.6 | 7.3 | 31.0 |
| Jagae-ri | Early LBA | *Oryza sativa* | D54-1 | 59.3 | 2.3 | −26.1 | 4.7 | 29.7 |
| Jagae-ri | Early LBA | *Oryza sativa* | D54-2 | 56.7 | 1.6 | −25.5 | 4.5 | 40.6 |
| Youngheung-ri | PTK | *Oryza sativa* | YHR-r1 | 60.4 | 1.7 | −25.7 | 4.1 | 40.9 |
| Youngheung-ri | PTK | *Oryza sativa* | YHR-r5 | 47.7 | 1.6 | −25.1 | 5.7 | 34.6 |
| Youngheung-ri | PTK | *Oryza sativa* | YHR-r6 | 58.3 | 1.8 | −23.9 | 7.7 | 36.9 |
| Youngheung-ri | PTK | *Oryza sativa* | YHR-r7 | 56.9 | 1.5 | −24.7 | 3.8 | 42.9 |
| Youngheung-ri | PTK | *Oryza sativa* | YHR-r8 | 51.1 | 2.1 | −24.0 | 5.7 | 28.9 |
| Youngheung-ri | PTK | *Oryza sativa* | YHR-r9 | 57.9 | 1.5 | −24.8 | 4.9 | 46.4 |
| Youngheung-ri | PTK | *Oryza sativa* | YHR-r10 | 48.9 | 1.6 | −24.2 | 6.2 | 36.5 |

**Table 2.** *Cont.*

| Site | Age/Phase | Plant Species | Sample Number | %C | %N | δ¹³C (‰) | δ¹⁵N (‰) | C/N |
|---|---|---|---|---|---|---|---|---|
| Youngheung-ri | PTK | *Oryza sativa* | YHR-r11 | 58.4 | 1.7 | −25.4 | 5.3 | 41.0 |
| Youngheung-ri | PTK | *Vigna angularis* | YHR-a1 | 48.2 | 4.2 | −26.4 | 4.6 | 13.4 |
| Youngheung-ri | PTK | *Vigna angularis* | YHR-a2 | 53.5 | 3.9 | −26.2 | 1.2 | 16.2 |
| Youngheung-ri | PTK | *Vigna angularis* | YHR-a3 | 40.8 | 2.6 | −24.8 | 1.3 | 18.4 |
| Youngheung-ri | PTK | *Vigna angularis* | YHR-a4 | 48.1 | 3.8 | −25.3 | 1.1 | 14.6 |
| Youngheung-ri | PTK | *Vigna angularis* | YHR-a5 | 59.0 | 5.3 | −24.9 | 0.1 | 13.0 |
| Youngheung-ri | PTK | *Triticum aestivum* | YHR-w1 | 48.1 | 2.6 | −24.5 | 5.6 | 21.4 |
| Youngheung-ri | PTK | *Triticum aestivum* | YHR-w2 | 52.6 | 2.9 | −24.0 | 3.9 | 21.5 |
| Youngheung-ri | PTK | *Triticum aestivum* | YHR-w3 | 56.2 | 3.1 | −24.4 | 2.4 | 21.0 |
| Youngheung-ri | PTK | *Triticum aestivum* | YHR-w4 | 57.6 | 3.1 | −23.0 | 3.5 | 21.8 |
| Youngheung-ri | PTK | *Triticum aestivum* | YHR-w5 | 54.7 | 3.5 | −23.7 | 3.4 | 18.3 |
| Simpo-ri | PTK | *Oryza sativa* | SPR-r1 | 57.6 | 2.0 | −24.2 | 3.3 | 33.6 |

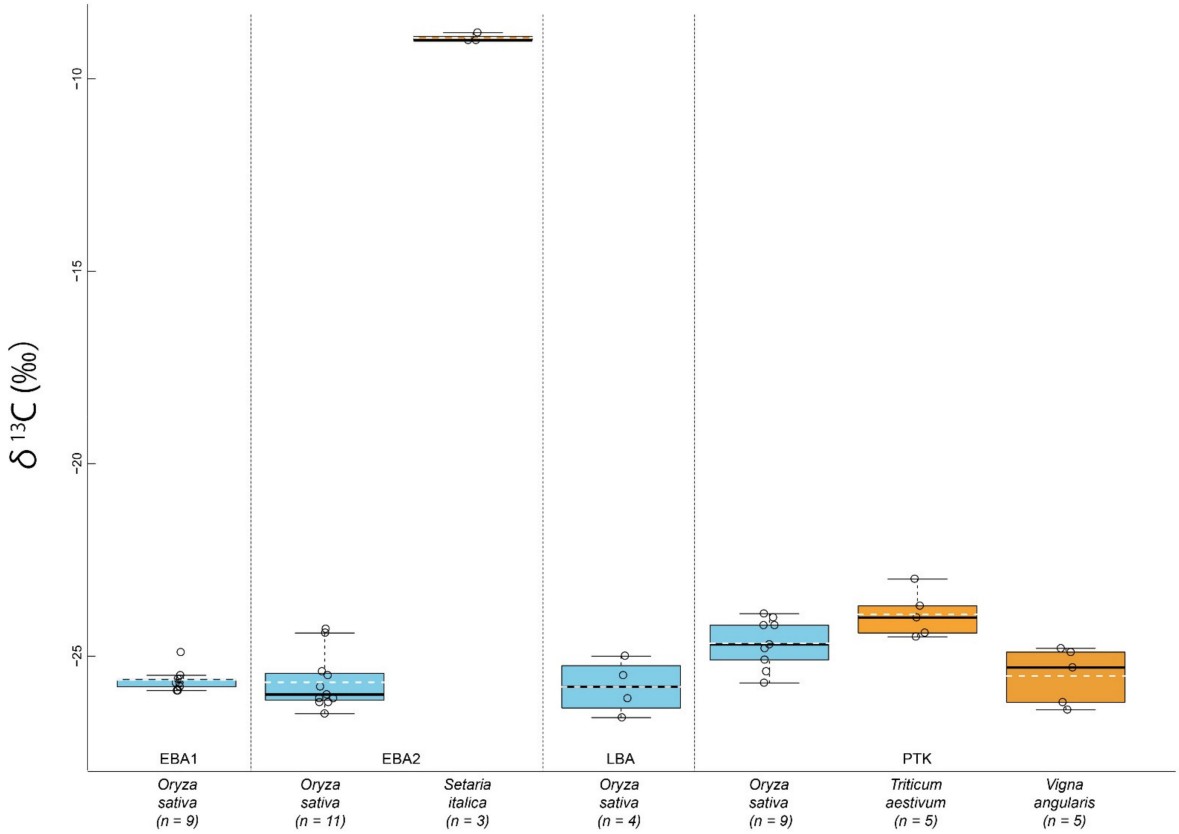

**Figure 3.** Boxplot comparing carbon stable isotope values among rice, wheat, and azuki bean in each period. Lower and upper box boundaries are 25th and 75th percentiles, error bars are upper extreme and lower extreme, respectively, the lines inside the box are the median, and broken lines are the mean value.

### 3.2. $\delta^{15}N$ in Rice and Other Crop Grains

Given the archaeological evidence of paddy fields found in the PTK period [52], elevated $\delta^{15}N$ in charred rice grains was predicted. As expected, significantly higher values (5.4 ± 1.2‰) were observed in charred rice than those in wheat from the same dwelling pit at the Yongheung-ri site (3.8 ± 1.2‰; *t*-test, *p* = 0.0342, Figure 4). From the same pit, azuki beans were also recovered, which had lower $\delta^{15}N$ values (1.7 ± 1.7‰) than those in wheat (*t*-test, *p* = 0.0533) and rice (*t*-test, *p* = 0.0007), which was quite reasonable because this plant incorporates atmospheric nitrogen, including less $^{15}N$ via the $N_2$ fixation, by the rhizobium [53,54]. According to this comparison, the biogenic isotope signatures of nitrogen have been kept in the archaeological materials.

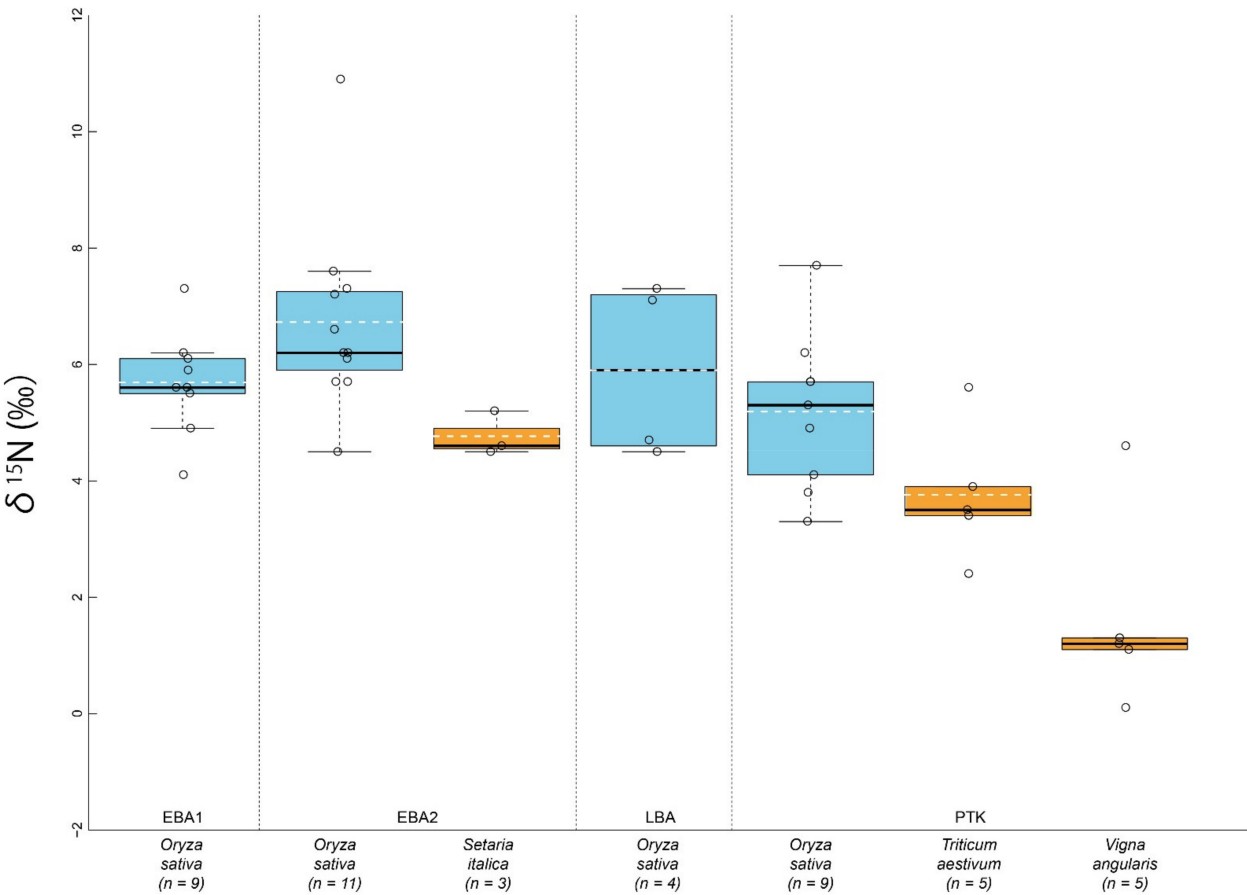

**Figure 4.** Boxplot comparing nitrogen stable isotope values among rice, wheat, and azuki bean in each period. Lower and upper box boundaries are 25th and 75th percentiles, error bars are upper extreme and lower extreme, respectively, the lines inside the box are the median, and broken lines are the mean value.

The higher nitrogen isotope values in rice were observed in all periods (Figure 4). It is reasonable that the $\delta^{15}N$ values of rice in the LBA (5.9 ± 1.5‰) had similar values to those of the PTK period (5.2 ± 1.3‰), given that some paddy fields excavated from the LBA were an indication of the continuity of wet rice cultivation from this age. Interestingly, the rice in EBA1 and 2 also had higher $\delta^{15}N$ values (5.7 ± 0.9‰ and 6.7 ± 1.6‰, respectively) than dry crops (Figure 4). A priori comparisons showed no significant difference in the $\delta^{15}N$ values in wet rice of the later periods (LBA + PTK) compared to those from the EBA1 (*t*-test, *p* = 0.2148) and EBA2 (U-test, *p* = 0.05504; EBA2 is higher than LBA + PTK). These data did not suggest the possible shift in $\delta^{15}N$ related to the transition of rice agriculture from dry (probably slash-and-burn) to wet as has been hypothesized in previous studies [7,25]. Moreover, the significant difference between the $\delta^{15}N$ values in rice (6.3 ± 0.9‰) and

millet ($4.8 \pm 0.4$‰) at the Baekseok-dong site of EBA2 (F-test, $p = 0.3066$; $t$-test, $p = 0.0183$; Figure 4) rather contradicted this hypothesis.

On the other hand, the $\delta^{15}$N values of foxtail millet of EBA2 and wheat of PTK were compared in order to investigate the conditions within a dry field. These values do not show a significant difference (F-test, $p = 0.1948$; $t$-test, $p = 0.208$).

## 4. Discussion

In this study, $\delta^{15}$N values of archaeological rice grains between the EBA and LBA/PTK were compared. Independent lines of archaeological evidence suggested that rice cultivation in the latter periods was carried out within a wet environment. The analysis done in the current study had no statistical support for a temporal change in the $\delta^{15}$N values of rice. Based on these data, there was no evidence that nitrogen cycles in both wet and dry fields changed drastically during these periods.

Additionally, the small variation in $\delta^{13}$C between species might reflect differences in water stress or growing conditions among crops [38,55]. It is noteworthy that the biological difference between species was not masked by the diagenetic alteration that would homogenize isotopic values in general. In addition, the $\delta^{15}$N values of rice were higher than those of millet recovered from the same context at both the Baekseok-dong and Yongheung-ri sites. Thus, these two crops were probably cultivated within different environments, such as wet and dry conditions.

Whilst higher $\delta^{15}$N values of wet rice than dry crops were ascribed to denitrification in paddy soils, it must be noted that there were some other potential factors which could have influenced the $^{15}$N-enrichment of soils, such as slash-and-burn cultivation and animal manuring. Szpak discussed three factors related to human activities that may affect the $\delta^{15}$N values of soil: (1) animal fertilizers; (2) burning/shifting cultivation; and (3) tillage [8]. Since it is assumed that animal-delivered fertilizer use commenced at the end of the Goryeo era (dated to the fourteenth century AD) on the Korean peninsula [52], the influence of fertilizers on the $\delta^{15}$N values of grains was not considered in this study. Unlike neighboring China, animal husbandry developed much later in Korea. No domesticated animal was reported, except dogs, in the Bronze Age and only limited domesticated animal bones were recovered from the PTK period [56]. Our data showed little change in the nitrogen cycle in both wet and dry conditions, suggesting little or no effect from fertilizer in the given period of time.

Secondly, immediate $^{15}$N-enrichment in surface soils after wildfires in sub-alpine ecosystems was reported [57]. However, there is still no consistent pattern in all cases because of a lack of systematic correspondence between vegetation burning and foliar/soil $\delta^{15}$N [8]. Additionally, if the $\delta^{15}$N values of the analyzed rice grains were enriched by burning, rice would have been separately cultivated using the slash-and-burn method, while millet and other dry crops would have been cultivated in other environments, a situation which is unlikely to have happened.

While deeper soils tend to have enriched $\delta^{15}$N values, especially in the forest [58], there is little influence on plant $\delta^{15}$N values caused by the ploughing or tillage of agricultural fields [8,59]. Accordingly, the plowing method does not seem to influence the $\delta^{15}$N values of crop grains. Moreover, it is recognized that people started animal plowing using iron, which changed cultivation depths drastically, from the Three Kingdom Period [52], i.e., following the periods covered in this study. Before that, "knapped stone axes" were most probably used as cultivating tools in both the Neolithic [60] and the Bronze Age [7]. In all cases, it was difficult to explain the higher $\delta^{15}$N values by tillage from both the isotope and archaeological data.

However, paddy field features dating to the EBA have not been identified. As the majority of excavations in modern South Korea are developer based [61], topographically appropriate areas for reclaiming paddy fields are rarely excavated. Thus, without large-scale construction, paddy field features are unlikely to be identified unless they accompany

contemporaneous settlements or cemeteries. In general, dating archaeological paddy field features is very problematic since few artefacts have been recovered from them.

Currently, 60 sites have been identified as paddy fields. Of these, 25 sites have been dated to "the Bronze Age". While the evidence indicates that the cultivation of rice crops in the EBA was present in the archaeological record, evidence for paddy fields of this date is yet to be confirmed. This is due to a lack of diagnostic artefacts providing definitive site dating. However, while these sites probably do belong to the Bronze Age, only two of 25 sites are confirmed as LBA. For those reasons, researchers are reluctant to claim the "earliest" paddy field if dating the site to the LBA cannot be excluded. In other words, it is still difficult to say whether paddy fields were absent during the EBA.

It is also notable that rice could be cultivated without clear traces of systematic irrigation as with wet rice. In fact, one of the weed assemblage studies in Japan suggested that early rice cultivation could be practiced without defined paddy ridges or compartments [34]. In Japan, it has been hypothesized that people practiced slash-and-burn crop cultivation in the Jōmon period, before the adoption of wet rice agriculture [62]. However, as there is no clear evidence of rice or millet cultivation before the beginning of the Yayoi period, and this totally contradicts our hypothesis based on the archaeobotanical data: rice and millet cultivation were seemingly combined on the Korean peninsula then diffused into the adjacent Japanese islands [27,34].

## 5. Conclusions

The early rice cultivation in the EBA was carried out in similar conditions to the rice cultivation in the LBA and PTK periods, which was probably carried out in wet conditions. The carbon and nitrogen stable isotope values of charred rice did not change drastically over a period of 1500 years. In addition, rice had higher $\delta^{15}N$ values than the other sampled species of dry crops during the EBA and the PTK periods. Therefore, rice was introduced and cultivated as a wet crop from the initial stages, rather than the slash-and-burn shifting cultivation, or dry field cultivation, which has been previously advocated without any concrete evidence. Although there are still several possible interpretations of the variety of $\delta^{15}N$ values, this paper hypothesized that the bacteriologic denitrification in anaerobic paddy soil caused the higher values in rice grains than other dry crops for better understanding of cultivation methods of rice in the past. The results successfully illustrated the wet cultivation of rice over a prolonged period of time, which was difficult to reconstruct from the available archaeological lines of evidence. Further application of this method to the adjacent areas will undoubtedly help us to understand the initial diffusion process of rice agriculture in East Asia.

**Author Contributions:** Conceptualization, S.S., S.-M.A., and M.Y.; Methodology, S.S., H.N., and M.Y.; Investigation, S.S., H.N., K.Y., M.Y.; Resources, S.-M.A. and G.-J.N.; Data curation, S.S., N.M., and M.Y.; Writing—original draft preparation, S.S. and M.Y.; Writing—review & editing, S.S.; Visualization, N.M.; Supervision, S.S.; Project administration, S.S.; Funding acquisition, S.S. and M.Y. All authors have read and agreed to the published version of the manuscript.

**Funding:** This research is supported by JSPS KAKENHI Grants-in-Aid for Young Scientists A (23682012, Grants-in-Aid for Scientific Research on Innovative Area (15H05969), Grant-in-Aid for Transformative Research Areas (20H05820), Grant-in-Aid for Scientific Research A (21H04370) and Marie Curie International Incoming Fellowship (II7-624467).

**Data Availability Statement:** The data presented in this study are available in this article and Appendix A.

**Acknowledgments:** We thank Lee S.-J., Itahashi Y., Robson H., Omori T., Vaiglova P., Craig O., Lucquin A., Colonese A., Hausmann N., Son J.-H., and Sugiyama S. for supporting this study. We also thank the anonymous reviewers for providing useful comments to improve this paper.

**Conflicts of Interest:** The authors declare no conflict of interest.

**Appendix A**

*Site Information*

1. The Gyo-dong site (37°45′ N, 128°53′ E) is located in Gangneung City, Gangwon-do Province, where residential land development led to the excavation. One Early Bronze Age dwelling pit (no. 1) was excavated in which plenty of charred rice grains were recovered from within the floor. (Baek, H-K., Ji, H-B. and Park, Y-K., 2002. *Report on the Excavation of Gyo-dong site in Gangneung.* Gangneun: Gangneun University Museum (in Korean).)

2. The Sacheon-ri site (38°20′ N, 128°14′ E) is located in Goseong County, Gangwon-do Province, where railway construction led to the excavation. Two Early Bronze Age dwelling pits were excavated. Charred rice was directly collected from the dwellings (no. C-5 and C-8). The former was analyzed in this study. (Yi, C-H., Ra, K-H., Kim, J-H., Sin, Y-R. and No, H-J., 2002. *Report on the Excavation of Sacheon-ri site in Goseong.* Chuncheon: Gangwon Research Institute of Cultural Properties (in Korean).)

3. The Baekseok-dong site (36°86′ N, 127°12′ E) is located in Cheonan City, South Chungcheong Province, where residential land development led to the excavation, which unearthed as many as 73 dwelling pits belonging to the Early Bronze Age. Together, with previously excavated settlements, this site is considered to be one of the most large-scale settlement sites in the Korean Bronze Age. Charred seed remains were collected by water flotation, which was undertaken on the soil from a dwelling pit (no. II-10). (Oh, G-J., Yi, P-S., Bae, S-H., Ahn, S-T. and Choi, K-S., 2009. *Report on the Excavation of Baekseok-dong Gojaemigol site in Cheonan.* Gongju: Chungcheong Institute of Cultural Heritage (in Korean).)

4. The Jagae-ri site (36°79′ N, 126°67′ E) is located in Dangjin City, South Chungcheong province, where highway construction led to the excavation, which resulted in a Bronze Age settlement consisting of nine dwelling pits. Seed remains were systematically collected by flotation. Charred remains from the soil taken from dwelling pits (no. 48 and 54) were used for the analysis in this study. (Na, G-J., 2006. *Report on the Excavation of Jagae-ri site in Dangjin.* Gongju: Chungcheong Institute of Cultural Heritage (in Korean).)

5. The Yongheung-ri site (35°53′ N, 127°09′ E) is located in Wanju County, North Jeolla Province, which was excavated as a developer-led project because of highway construction. Rice grains were collected from sediment recovered from the floor of a dwelling pit (no.1). This dwelling pit is one of 14 belonging to the Proto-Three Kingdoms period at this site, together with other cereal grains such as millet and bean, as well as charcoal, parts of which were analyzed in this study. (Kim, G-J, Yangm Y-S and Ahn, H-J., 2008. *Report on the Excavation of Yongheung-ri in Wanju, Jeollabuk-do.* Imsil: Jeonbuk Cultural Property Research Institute (in Korean).)

6. The Simpo-ri site (35°85′ N, 126°71′ E) is located in Gimje City, North Jeolla Province. A dwelling pit belonging to the Proto-Three Kingdoms period was found due to an academic general survey. Plenty of rice grains were excavated from the pit with burned soil in the dwelling pit. (Kim, J-K and Na, W-S., 1999. On the Charred remains excavated from the Simpo-ri site in Gimje. *Report on the General survey of the coastal area in Buan Cheonan.* Jeonju: Jeonju National Museum (in Korean).)

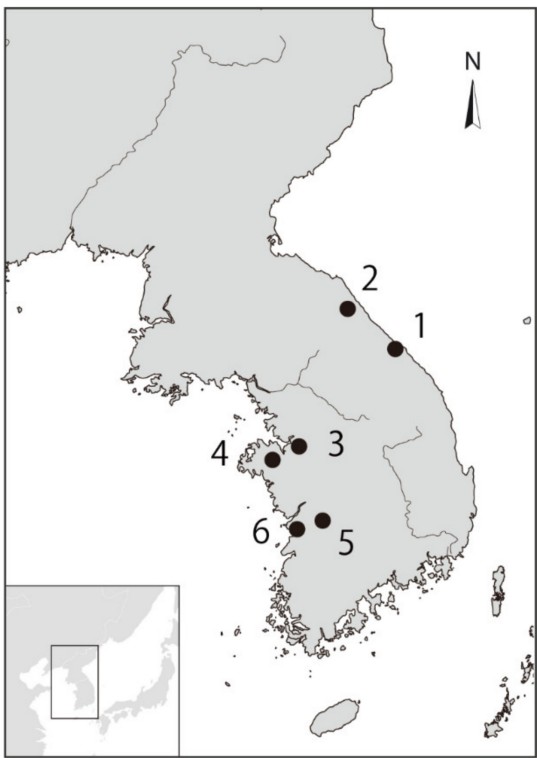

**Figure A1.** Locations of the archaeological sites investigated in this paper (1: Gyo-dong, 2: Sacheon-ri, 3: Baekseok-dong, 4: Jagae-ri, 5: Yougheung-ri, 6: Simpo-ri).

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
