# Peer review of "Dry or Wet? Evaluating the Initial Rice Cultivation Environment on the Korean Peninsula"

_agronomy, doi:10.3390/agronomy11050929_

Round 1

Reviewer 1 Report

Is it possible that an advanced draft and not the final version was submitted? There remains some track changes and undeleted advice. Also the English could be improved. For example, the first sentence of the abstract “The origins and development of rice cultivation is one of the most important domestications in East Asian prehistory” doesn’t really make sense - not off to a good start. Also, my preference is to use BCE or BP rather than BC, but that is a minor thing. It would be good to be consistent in the paper. Some mishmash of form and from which isn't picked up by spell checking. Spell checking still needed though.

Methods are adequately described, though I think more discussion of the geography and EBA/LBA climate of the area is needed, particularly given you are comparing wet and dry agriculture.

The crux of the paper lies in the natural abundance N15/N14 analysis and interpretation, and this made interesting reading. The argument for the increased N15 being mostly due to nitrogen cycling in wet soil should be expanded. Differential livestock manuring compared to classically upland crops, even through greater livestock grazing, could conceivably drive up the relative natural abundance of N15. This is discussed a little already, and the rationale for disregarding potential manuring could be strengthened, as it seems like a feasible alternative explanation.

Is it possible to include a little more explanantion about what you mean by wet rice in the ancient context. Whilst we tend to think of made rice paddies, of course there would have been naturally wet areas in which rice could be cultivated. There is a temptation to be wildly speculative, so just some mention of non-paddy wet rice would be sensible, perhaps with some examples (at the extreme there is deepwatwer/floating rice on some deltas, though I guess thats not applicable here). Or if you think thats not likely or possible in this specific location would be helpfully argued. 

Reviewer 2 Report

The authors present a data set examining nitrogen isotopes in ancient grains in Korea, with the aim of demonstrating no shift in isotopic values and suggesting no change in rice cultivation systems over time. 

The most significant issue is that the sites are not adequately accounted for in the data analysis. Since the time periods are represented by different sites, there could be confounding factors (in space) that influence the patterns described (in time). At the bare minimum, the same time-series analysis of the non-rice grains needs to be shown.  Related to that, the sites are not described and the Appendix of the studies was not attached. 

Overall, the submission seems very rushed.  Lots of gramatical errors.  "Slush-and-burn."  Entire letters missing.  Still have comments and didn't accept all the changes in the document.  Any resubmission would need careful copyediting. 

Methods were fairly thin. How were the grains identified?  Were there any that were ambiguous?  Or thrown out for any reason?

Under Methods, the authors state "Considering that ABA pre-treated samples tend to have higher δ15N values than non-treated ones [41], we did not apply alkaline washing."  Was this for all samples, or were some samples treated differently than others?  If alkaline washing is the method, why forgo it?  Yes it changes the 15N values but isn't that part of the cleaning process? Related to that, ABA is never defined anywhere in the manuscript. 

Overall the presentation of the results could be made much more clearly.  Seems like 2 or so graphs could capture all the results in a very efficient way. 

Reviewer 3 Report

Dear authors, Could you kindly bring out the rational of your study. Kindly tell us the adding value of your study to wet and dry rice production in study area in particular and in the world if possible.Comments are in the pdf document received. Thanks

Round 2

Reviewer 2 Report

The authors have addressed all comments raised in my previous review. 

Author Response

Thank you very much for taking the time for the second review.

I have collected spelling issue (typically analy"z"e and analy"s"e) in the text.

Shinya

Reviewer 3 Report

Dear author,

Kindly find comments to incorporate to your manuscript for it improvement.

Line 49, write 'to understand'; L68, write 'remains dating were analyzed to ...'; L96, delete 'this paper also stated that' and write 'Additionally, the majority...'; L132, replace 'are' by 'were'; L141, put '.' after 'mobility' and start the next sentence with 'A'; L146, delete 'the other is that' and replace with 'Also, there is...'; L161, delete 'when'; L178, delete 'Also,'; L180, delete 'that have been'; L201, delete 'by this' and replace by 'using the mentioned method'; L226, delete 'are' in 'grain are described' and replace with 'were'; From L226 to L241, take some results stated to the results section; L256, write 'EBA was divided ...'; L270, delete 'Firstly,'; L275, write 'alkaline washing was not applied for ...'; L292, write 'non-parametric methods were applied...'; L297, delete 'Figure 4'; L298, write 'minor differences were found ...'; L304, replace 'is' with 'was'; L321, delete 'figure 4'; L328, replace 'is' with 'was'; L368, delete 'do' and write 'did'; L376, delete 'As a result,'; L279, what is that figure for? where is it stated in the text? what is the title? take it out if not necessary.; L405, delete 'were'; L411, delete 'Firstly,'; L419, write 'in the given period of time' instead of 'in the given time period'; L428, delete 'Thirdly,'; L460, replace 'have' with 'is'; L463, write 'confirmed as LBA ...'; L495, change 'causes' into 'caused'; delete 'compared to ...' and replace by 'than ...'; L496, replace 'the cultivation ...' by 'of cultivation ...'; replace 'illustrate' by 'illustrated'; L491, delete 'these results support the interpretation that' and replace by 'Therefore, rice was ...'.

I did not attach any additional document to this message

Best regards

Author Response

Thank you very much for reading through the manuscript again.

So much appreciated for using your precious time on brushing up on our work. 

All the points were corrected following the suggestion except below;

L226 -241 I understand what the reviewer means but we would rather this part there as this is not the main part of the analysis of this paper (such as AMS dating in the same chapter).

L304 I could not find "with" to delete in this line.

L405 If I delete "were" from here it seems grammatically erroneous.